# The causal effect of stressors on antenatal common mental disorders among pregnant mothers in Dabat district Northwest, Ethiopia: A generalized structural equation model approach

Helina Abebe Kurbi[1,2]*, Solomon Mekonnen Abebe[3], Netsanet Worku Mengistu[3], Alemayehu Teklu Toni[4], Tadesse Awoke Ayele[1]

1 Department of Epidemiology and Biostatistics, Institute of Public Health, College of Medicine and Health Sciences, University of Gondar, Gondar, Ethiopia, 2 Menelik II Health Science College, Department of Mental Health, Addis Ababa, Ethiopia, 3 Department of Nutrition, Institute of Public Health, College of Medicine and Health Sciences, University of Gondar, Gondar, Ethiopia, 4 Department of Pediatrics and Child Health, School of Medicine, University of Gondar, Gondar, Ethiopia

* helusimren@gmail.com

**Data availability statement:** The University of Gondar's ethical review committee has restricted the sharing of this dataset due to the sensitive nature of the study. However, data supporting the findings can be made available upon reasonable request to researchers who meet the ethical and legal criteria. The corresponding

## Abstract

Antenatal mental health is a crucial but overlooked and neglected element of maternal and infant health care. Various factors have been associated with triggering mental illness during pregnancy. It is essential to explore the stressors and mediators that play a role in causing these health issues and understand how they interact. Therefore, this study aimed to analyze the causal association between stressors and factors influencing mental disorders using the stress process model. A community-based cross-sectional survey was conducted among pregnant women at Dabat Health and Demographic Surveillance Site Northwest, Ethiopia, in June and August 2022. This study aimed to measure common mental disorders in pregnant women using Self-Reported Questionnaire-20 and explore the causal association with stressors variables. A Generalized Structural Equation Model was used to evaluate the effect of stressors and mediators on common antenatal mental disorders. This model, known for its ability to understand the complex relationship between environmental exposures, emotional responses, and cognitive appraisals, was used to evaluate the impact of stressors and mediators on common antenatal mental disorders. The study included 872 pregnant women, of which 114 (13.1%) (95% CI: 10.07, 15.4) reported experiencing symptoms of common antenatal mental disorders. Lack of antenatal care($\beta$=1.4), unplanned pregnancy($\beta$=0.85), and intimate partner violence ($\beta$=0.87), all of which were associated with a higher antenatal common mental disorder score. A family history of mental disorders ($\beta$=-0.11) was associated with a decreased antenatal common mental disorder score. Moreover, social support ($\beta$=-0.19), life-threatening events ($\beta$=0.93), and stress-coping mechanisms ($\beta$=0.12) appeared to mediate the link between the identified stressors and antenatal common mental disorders. This study highlights the association between psychosocial factors and antenatal mental health, underlining the urgent need for a holistic approach to prenatal care that includes mental well-being.

author will evaluate each request carefully in consultation with the ethical review committee and the Dabat Research Center.

**Funding:** The University of Gondar funded this study's data collection and materials costs; however, the funder had no say in the study design, analysis, manuscript preparation, or publication decision.

**Competing interests:** The authors have declared that no competing interests exist.

**Abbreviations:** MD, Common Mental Disorder; ANC, Antenatal care; HDSS, Health, and Demographic Surveillance Site; SRQ-20, Self-Reporting Questionnaire 20; LTE, Life-Threatening Experiences; ODK, Open Data Collection Kit; OSSS, Oslo Social Support Scale; GSEM, Generalized Structural Equation Method; PCL-4, Perinatal Coping Inventory; TLI, Tucker Lewis Index; CFI, Comparative Fit Index; SRMR, Standardized Root Mean Square Residuals.

Integrating support systems and stress-coping mechanisms is crucial for improving maternal health outcomes.

## Introduction

Common mental disorders (CMD), such as depression, anxiety, and somatoform disorders, impose a significant public health burden globally [1]. Studies indicate that the prevalence of common mental disorders varies significantly, affecting approximately 10% of women in high-income countries. In comparison, nearly 20% of women in low and middle-income countries, particularly in Sub-Saharan Africa and South Asia, face these challenges [2,3]. These global estimates highlight the disproportionate burden faced by women in low-resource settings like Ethiopia, where socioeconomic, cultural, and healthcare-related factors exacerbate the risk of mental disorders during pregnancy[3–6]. The urgency of addressing this issue cannot be overstated.

In Ethiopia, the prevalence of common antenatal mental disorders poses a significant public health issue, with prevalence rates ranging from 12% in Butajira Health and Demographic Surveillance Site [7] to 37.5% in Kersa and Haramaya Health and Demographic Surveillance sites (HDSS) [8]. These variations can be linked to a range of socioeconomic and cultural factors. The difference in the prevalence of common mental disorders during the perinatal period is influenced by different factors such as poverty, food insecurity, unemployment, and socioeconomic and cultural factors [9], which hinder access to health care[10]. Furthermore, cultural perception and societal norms contribute to the stigma surrounding perinatal mental disorders, resulting in underreporting [11–13].

Common antenatal mental disorders can be influenced by factors including experiencing violence from a partner, having an unplanned pregnancy, facing financial difficulties, having a family history of mental illness, previous miscarriages, and food insecurity [14–17]. These factors can lead to absent prenatal care follow-up, delayed breastfeeding, longer labor duration, and premature birth. Additionally, they can impact the child's growth and development regarding behavior, movement skills, and thinking abilities [3,18–22].

Mental health care for mothers remains an overlooked aspect of maternal and child care [23]. Maternal mental disorders have become a significant but often neglected burden in low- and middle-income countries, including Ethiopia[24]. In Ethiopia, pregnant women face different stressors, such as poverty, limited access to health care, cultural pressures, social instability [25], lack of screening, scarcity of mental health providers, and stigma [26], which can contribute to the presence of common mental disorders.

Therefore, this study aims to fill the gap by investigating the causal associations of stressors and mediators of antenatal CMD among pregnant women in Ethiopia. The research uses the well-established Pearlin et al. stress process model as a theoretical framework to guide our analysis. This model offers a comprehensive framework encompassing stressors, mediators, and stress outcomes, providing a clear structure for our study. Our research's extensive nature ensures our findings' validity and reliability.

Generalized Structural Equation Modeling (GSEM) explored the causal mechanism between stressors, mediators, and common antenatal mental disorders. GSEM handles complex relationships and diverse data types like antenatal CMD [27]. This rigorous methodology has the strength to study to examine direct and indirect effects in detail. Conducting this research in the Ethiopian context is particularly important due to the unique socio-economic, obstetric, and psychosocial characteristics influencing maternal mental health, which can all contribute to the development of antenatal CMD. This study aims to examine the causal

effects of various stressors on antenatal CMD, assess the mediating role of social support, and evaluate the indirect effects of stressors of antenatal common mental disorders through mediators. The findings of the study not only help to understand the causal association of maternal mental health in low- and middle-income countries but also inspire hope for future research and interventions.

## Methods

A community-based cross-sectional study was conducted among pregnant women residing in the Dabat district Northwest, Ethiopia, from June 1 to August 30, 2022.

### Ethics statement

This study has been reviewed and approved by the Institutional Review Board of the University of Gondar under the reference number VP/RTT/Eng./051/06/2021. The relevant health authorities and Dabat Research Center have also approved and issued a letter of approval. The study was conducted following the Declaration of Helsinki. Before participating, all participants were given adequate information about the research and its potential benefits and risks, and informed verbal consent was obtained from all mothers. Mental health professionals assessed participants who showed significant levels of symptoms of CMD. If necessary, a referral to Gondar University Hospital was provided.

### Study setting and participants

The research was conducted at the Dabat Health and Demographic Surveillance Site (HDSS) in Dabat District Northwest, Ethiopia. The HDSS site, a significant research center, covers nine rural and four urban randomly selected kebeles. Dabat is one of the 21 districts situated in the North Gondar Administrative Zone of the Amhara Region in Ethiopia. Dabat town, the capital of Dabat District, is located approximately 821 km northwest of Addis Ababa and 75 km north of Gondar. The HDSS has collected data on vital events like birth, death, migration, and pregnancy registrations and outcomes quarterly and regularly. The Dabat HDSS is a full member of the International Network of Demographic Evaluation of Population and Their Health (INDEPTH) [28].

**Study population.** The source population for this study included all pregnant women living in the Dabat HDSS. All registered pregnant women in their second or third trimester were included. As these stages are characterized by significant emotional and physical transformations due to hormonal fluctuations, it is critical to investigate their impact on mental well-being and ensure comprehensive representation. Pregnant women who could not communicate due to severe medical or psychiatric conditions were not included in the study. Participants were identified using the Dabat HDSS database and local health extension antenatal registration records and interviewed at their homes.

### Sample size determination

This research is part of a large community-based prospective cohort study to investigate the impact of common perinatal mental disorders (CMDs) on low birth weight, infant growth, and development in the Dabat Health and Demographic Surveillance System (HDSS) in Northwest Ethiopia [29]. The sample size calculation was estimated based on the estimated effect of antenatal depression on low birthweight[30]. The sample size was calculated using Epi-info version 7 [31]. The double population formula was used with the assumption of a 95% level of confidence, 5% margin of error, 90% power, exposed to non-exposed ratio of 1:2, and the prevalence of low birth weight among those free from antenatal depression of 21%,

an effect size of 1.5, and 10% loss to follow-up; as a result, the final sample size was estimated 946. Proportional allocation of the sample size based on the estimated number of pregnant women in each Kebele within the Dabat HDSS. Then, systematic random sampling was used to select participants, and the sampling interval (Kth interval) was determined by dividing the estimated number of pregnant women by the allocated sample size.

**Data collection methods.** A comprehensive and pre-tested electronic survey was conducted in person with pregnant women, utilizing the Open Data Collection Kit (ODK) application designed to collect, manage, and use data in low-income settings [32].

This tool enhances data accuracy through its built-in features. ODK allows real-time data verification by applying restrictions on acceptable values and checking for logical consistency. It also implements skip logic and conditional branching, ensuring that questions relevant to their previous answers are asked, which reduces the risk of errors. Moreover, ODK automatically captures metadata, including time stamps and GPS coordinates, bolstering data integrity by removing the need for manual entry and offering immediate feedback on possible mistakes. As a result, ODK facilitates gathering high-quality, dependable data crucial for the credibility of research findings [33].

A team of 28 data collectors and seven supervisors were trained in the local language, Amharic, for three days. They collected data at the Dabat HDSS site and sent completed forms through a wireless connection. Local informants were crucial in the field, promptly reporting pregnancies and births. Data quality was a top priority, ensured through on-site supervision and random rechecking of the respondents in each locality.

## Measures

Antenatal common mental disorders were measured during the second and third trimesters of pregnancy using the locally validated Self-Reporting questionnaire (SRQ-20) [34]. The SRQ-20 questionnaire consists of 20 items that help evaluate the presence of depressive, anxiety, panic, and somatic symptoms in the preceding 30 days. It generates a scale indicating the overall level of psychological distress. It has been validated for use in pregnant and postnatal women in the Butajira population [35,36]. Also, the tool has shown good cultural adaptation, validity, and reliability in Northwest Ethiopia.

The Stressful Life: List of Threatening Experience (LTE-12) questionnaire was translated into Amharic and adapted for local conditions and time frames restricted to the current pregnancy. The scale contains 12 items, including questions about death, illness, conflict, and property loss. It has good test-retest reliability (Kappa=0.61-0.87) and predictive validity [37]. The OSSS-3 categorizes social support as poor, moderate, or strong, depending on the score range of 3-8, 9-11,] and 12-14, respectively. It has good convergent and predictive validity [38].

A woman's stress coping level was assessed using the four-level customized internally consistent coping subscales of the perinatal coping inventory (PCL-4), which was explicitly developed for pregnancy [39]. The WHO Multicounty Study Questionnaire was used to assess intimate partner violence, consisting of psychological, physical, and sexual violence. A positive answer to any of these types of violence indicated the presence of violence [40].

## Data management and analysis

Completed data were downloaded from the Dabat HDSS server in Excel spreadsheet format, checked for completeness, and imported to STATA version 16(Stata Corp, USA) for further cleaning and analysis. Descriptive statistics such as mean, median, standard deviation, and percentage were used to summarize the data appropriately. The chi-squared test was used to test for crude associations between the categorical stressors and evidence of antenatal CMD.

The Hosmer–Lemeshow statistics tested the model's goodness of fit. Multicollinearity was assessed using the variance inflation factor (VIF) for each predictor in the binary logistic regression model. All VIF values were below the threshold of 5, with the higher value being 1.10, indicating no significant multicollinearity among the independent variables.

A Generalized Structural Equation Modeling (GSEM) was constructed to reflect the stress–process model framework and explain the direct and indirect relationships between the stressors and outcome variables to assess the strength of the hypothesized causal indirect and direct pathways. Prior subject knowledge and multivariable binary logistic results were used to select the potential stressors and causal paths. Before performing the GSEM analysis, only statistically significant predictors (p<0.05) from the multivariable binary logistic regression analysis (S1 Table) were used to build up the GSEM model [41,42]. The model was iteratively modified by adding or removing paths until a theoretically and statistically sound model was achieved.

The GSEM approach was preferred because it can handle discrete and continuous endogenous variables. Given that CMD responses are typically expressed as binary data (yes/no), the traditional SEM method is not suitable for exploring the causal pathway. Therefore, we employed a new path analysis called generalized SEM (GSEM), which can estimate maximum likelihood with a logit link function for binary outcome variables; GSEM integrates structural equation modeling (SEM) with generalized linear models (GLMs) which allows the inclusion of stressors (independent variables) and mediators in the same framework [43,44].

## Results

Out of the 888 pregnant women contacted in the Dabat HDSS, 10 refused to consent to the study, and six were unavailable after three further attempts to contact them. As a result, 872 pregnant women agreed to participate in the survey, resulting in a response rate of 98.1%. The mean age of the pregnant women was 27.6± 6.1SD. Most participants, 95.4%, were followers of Orthodox Christianity. Housewives accounted for 84% of the pregnant women, and 58.7% reported having no literacy skills (Table 1).

### Maternal and obstetric characteristics of pregnant women

The maternal and obstetric characteristics show that most (74%) pregnancies were planned, and most (63%) were in their third trimester. About 50.3% of pregnant women did not start antenatal care follow-up, the history of abortion was 6.77%, and the history of stillbirth accounts for 5.39% among pregnant women. The proportion of CMD was high (20.33%) among pregnant women having a history of abortion (Table 2).

### Psychosocial characteristics of pregnant women

Fig 1 displays the Psychosocial characterstics of the participants. The majority, 94.61%, of participants had no family history of psychiatric illness. Compared to non-CMD participants, a higher percentage of women in the CMD group reported a family history of Psychiatric illness (p = 0.000). 86.12% of participants reported strong social support, and 56.65 reported strong support from their partners. Furthermore, 39.3% of the participants reported experiencing intimate partner violence Fig 1.

### Effect of Stressors on Common Mental Disorder

Our GSEM analysis on the effects of stressors and mediators on antenatal CMD scores revealed significant findings, as shown in Table 3 and Fig. 2. Lack of antenatal follow-up has a substantial direct effect β=1.4) on CMD scores and the negative indirect effect (β= 0.15) suggests that mediating factors may mitigate the indirect impacts in reducing total effects (β=

**Table 1. The Socio-Demographic Characteristics of Pregnant Women.**

| Variables | ANC CMD YES (no, row%, column%) | ANC CMD NO (no, row%, column%) | Total (no, row%, column%) | P-value |
|---|---|---|---|---|
| Age | | | | 0.499 |
| 18-25 | 48(13.71, 42.11) | 302(86.29,39.84) | 350(100, 40.14) | |
| 26-35 | 55(13.06, 48.25 | 366(86.94,48.28) | 421(100, 48.28) | |
| Greater than 35 | 11(10.89, 9.65) | 90(89.11,11.87) | 101(100, 11.58 | |
| Residence | | | | 0.849 |
| Urban | 46(12.81, 40.35) | 313(87.19, 41.29) | 359(100, 41.17) | |
| Rural | 68(13,25, 59.65) | 445(86.74, 58,71) | 513(100, 58.83) | |
| Marital Status | | | | 0.022 |
| Married | 99(14.43, 86.84) | 587(85.57, 77.44) | 686(100, 78.67) | |
| Others [a] | 15(8.06, 13.16) | 171(91.94, 22.56) | 186(100, 21.33) | |
| Religion | | | | 0.711 |
| Orthodox | 108(12.98,94.74) | 724(87.02, 95.51) | 832(100,95.41) | |
| Muslim | 6(15.00, 5.26) | 34(85.00, 4.49) | 40(100, 4.59) | |
| Mother's Education | | | | 0.385 |
| Unable to read and write | 64(13.13, 56.14) | 423(86.86,55.80) | 487(100, 55.85) | |
| Able to read and write | 18(11.11, 15.79) | 144(88.89,19.00) | 162(100, 18.58) | |
| Primary | 9(10,23, 7.89) | 79(89.77,10.42) | 88(100, 10.09) | |
| Secondary | 23(17.04, 20.18) | 112(82.96, 14.78) | 135(100, 15.48) | |
| Mother's Occupation | | | | 0.53 |
| Housewives | 95(12.93, 83.33) | 640(87.07, 84.43) | 735(100, 84.29) | |
| Government Employee | 14(16.28,12.28) | 72(83.72, 9.50) | 86(100, 9.86) | |
| Others [b] | 5(9.80, 4.39) | 46(90.20, 6.07) | 51(100, 5.85) | |
| Monthly Income in ETB | | | | 0.841 |
| Less Than 2000 ETB | 82(13.44, 71.93) | 528(86.56, 69.66) | 610(100,69.55) | |
| 2000-4000 ETB | 21(12.73, 18.42) | 144(82.27, 19.00) | 165(100, 18.92) | |
| Greater Than 4000 ETB | 11(11.34, 9.65) | 86(88.66, 11.35) | 97(100, 11.12) | |
| Families Living In the House | | | | 0.657 |
| One-Five Families | 73(13.47, 64.04) | 469(86.53, 61.87) | 542(100, 62.16) | |
| Greater Than Five | 41(12.42, 35.96) | 289(87.58, 38.13) | 330(100, 37.84) | |

[a]single, divorced, and widowed.

[b]daily laborer and private employee, ETB Ethiopian Birr.

0.90). Regarding the effects of unplanned pregnancies had both direct (β=0.85) and indirect effects (β=1.80) on CMD score, Intimate partner violence had both direct (β=0.87) and indirect effects (β=0.17), contributing to an increased CMD score. The importance of considering a family history of health psychiatric illness cannot be overstated, as it was associated with both adverse (β= -2.09) and indirect effects (β= = 0.60), resulting in a considerable negative total effect (β= -3.1) on antenatal mental health scores Table 3.

The effect of social support has a negative total effect (β=0.19) on antenatal CMD score, indicating that as the level of social support increased by one standard deviation from low to high, a mother estimated common mental disorder score by 0.19 total effect (β=0.19)was protective. At the same time, stress coping shows a positive direct impact of (β=1.2) on CMD scores, and life-threatening events had a direct effect (β=0.93) on CMD scores, indicating a significant impact on CMD scores. Our study comprehensively analyzes the factors affecting antenatal common mental disorder scores using the Pearlin stress Model and GSEM analysis Fig 2.

**Table 2. Obstetric and Clinical Characteristics of Pregnant Women.**

| Variables | ANC CMD YES (no, row%, column%) | ANC CMD NO (no, row%, column %) | Total (no, row%, column%) | P-value |
|---|---|---|---|---|
| **Number of Pregnancies** | | | | 0.285 |
| 0ne pregnancy | 23(17.29, 20.18) | 110(82.71, 15.51) | 133(100,15.25) | |
| 2-4 pregnancies | 55(12.09, 48.25) | 400(87.91, 52.77) | 455(100, 52.18) | |
| Five or More Pregnancies | 36(12.68, 31.58) | 248(87.32, 32.72) | 284(100, 32.57) | |
| **Pregnancy Intention** | | | | 0.000 |
| Planeed | 44(6.82, 38.60) | 601(93.18, 79.29) | 645(100,73.97) | |
| Unplanned | 70(30.84, 61.40) | 157(69.16, 20.71) | 227(100, 26.03 | |
| **History of Abortion** | | | | 0.086 |
| Yes | 12(20.34, 10.53) | 47(79.66, 6.20) | 59(100, 6.74) | |
| No | 102(12.55, 89.47) | 711(87.45, 93.80) | 813(100, 93.23) | |
| **Trimester** | | | | 0.114 |
| Second Trimester | 35(10.74, 30.70) | 291(89.26, 38.39) | 326(100, 37.39) | |
| Third Trimester | 79(14.47, 69.30) | 467(85.53, 61.61) | 576(100, 62.61) | |
| **History of Stillbirth** | | | | 0.773 |
| Yes | 5(14.71, 4.39) | 29(85.29, 3.83) | 34(100, 3.90) | |
| No | 109(13.01, 95.61) | 729(86.99, 96.17) | 838(100, 96.10) | |
| **ANC Follow-Up** | | | | 0.000 |
| Yes | 78(18.01, 68.42) | 355(81.99, 46.83) | 433(100, 49.66) | |
| No | 36(8.20, 31.58) | 403(91.80, 53.17) | 439(100, 50.34) | |
| **Nutritional Status of the Mother** | | | | 0.127 |
| Poor Nutrition | 33(10.71, 28.95) | 275(89.29, 36.28) | 308(100, 35.24) | |
| Good Nutrition | 81(14.36, 71.05) | 483(85.64, 63.72) | 564(100, 64.68) | |
| **History of Chronic Medical Illness** | | | | 0.017 |
| Yes | 20(20.83, 17.54) | 76(79.17, 10.03) | 96(100, 11.01) | |
| No | 94(12.11, 82.46) | 682(87.89, 89.97) | 776(100, 88.99) | |

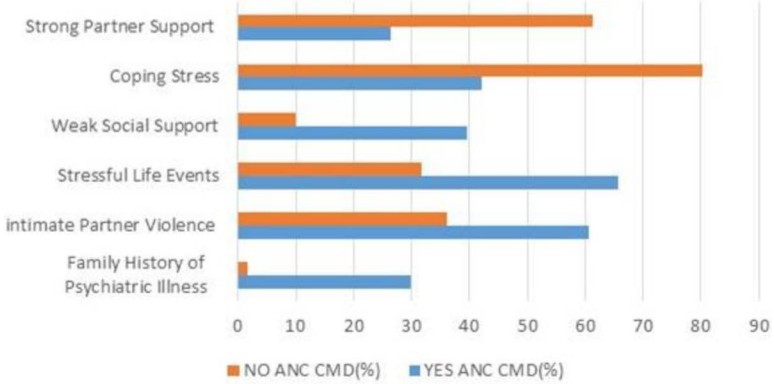

**Fig 1. Psychosocial characteristics of pregnant women in Northwest Ethiopia.** Prevalence of Common Mental Disorder This study revealed that 13.1% (95% CI: 10.7-15.4%) of pregnant women experience common mental disorders during the antenatal period. The SRQ-20 tool demonstrated tool high reliability (α=0.809) for measuring participants' CMD, with headache (31.1%) and loss of appetite (30.4%) being the most commonly reported symptoms.

**Table 3. Direct, Indirect, and Total Effects Stressors on Antenatal Common Mental Disorder.**

| Risk Factors | Direct Effect (β,se) | Indirect Effect(β,se) | Total Effect(β,se) |
|---|---|---|---|
| ANC Follow-Up | | | |
| Not started | 1.4(0.25)*** | -0.15(0.12) | 0.90(0.20)*** |
| Pregnancy intention | | | |
| Unplanned | 0.85(0.27)*** | 0.31(0.07)*** | 1.80(0.20)*** |
| Family history of psychiatric illness | | | |
| Yes | -2.09(0.51)*** | -0.60(0.15)*** | -3.1(0.37)*** |
| Intimate partner violence | | | |
| Yes | 0.87(0.26)** | 0.17(0.08)* | 1.00(0.24)*** |
| Life-threatening events | | | |
| Yes | 0.93(0.26)** | 0.00 | 0.93(0.26)** |
| Social support | | | |
| Strong | -0.19(0.40) | 0.00 | -0.19 |
| Coping stress | | | |
| Good | 1.2(0.27)*** | 0.00 | 1.2(0.27)*** |
| Partner support | | | |
| Yes | 1.1(0.02)*** | 0.39(0.08)*** | 3.15 (0.30)*** |

*p-value<0.005, **p-value <0.001, ***p-value<0.000. β= estimates, se=standard error

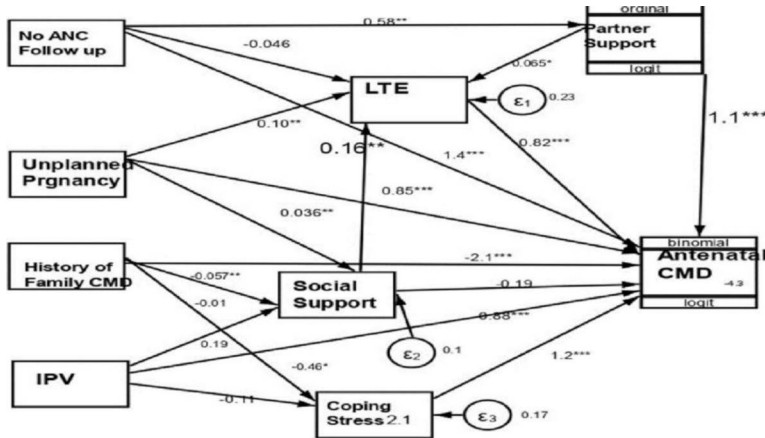

**Fig 2. Generalized Structural Equation Modeling for Effect of Stressors on Antenatal Common Mental Disorders.**

## Discussion

The study results highlight the complex nature of common antenatal mental disorders. Unplanned pregnancy, Family history of mental disorders, experiencing intimate partner violence, and lack of antenatal care follow-ups are factors significantly associated with common mental disorders among pregnant mothers. Additionally, the research revealed that social and partner support and coping strategies could mediate the causal association between common antenatal mental disorders and stressors. These findings emphasize the significance of a supportive environment and stress the need for a comprehensive approach and integrating

maternal mental health as an essential aspect of antenatal care, which includes maternal mental well-being, which is a crucial component of prenatal services.

This study found that 13.1% of pregnant women had an antenatal common mental disorder. This prevalence was slightly higher than the estimate for Butajira town (8%) [18], Arba Minch Zuria district (10.0) [45], and a study conducted in West Africa 7% [46]. Our estimates for the prevalence of common antenatal mental disorders were lower than previous estimates in other regions of Ethiopia [8,15,47,48] and other countries [6,49–53]. The variation in prevalence rates across different regions and countries underscores the influence of diverse factors such as cultural norms, healthcare accessibility, quality of maternal health services, and specific cultural practices. These factors can significantly impact the mental health of pregnant individuals. Additionally, variations in research methodologies, sample sizes, and the psychosocial conditions of the populations studied further complicate direct comparisons between studies. This complexity suggests a multifaceted approach is needed to understand and address antenatal mental health, considering each population's unique cultural, social, and healthcare contexts. It also highlights the importance of standardizing research methodologies to improve the comparability of studies in this field.

This study also found that a family history of mental disorders was associated with common maternal mental disorders during pregnancy, both directly and indirectly. This association suggests that there might be a genetic predisposition as stress plays a role in influencing the symptoms experienced during pregnancy. Additionally, a family history of mental disorders appeared to be influenced by social and stress-coping mechanisms. This implies that shared environmental factors, learned behaviors, and genetic susceptibilities within the family interact with environmental stressors, contributing to the complexity of mental well-being in expectant mothers [54]. Understanding this interplay provides valuable insight into the nature of mental health challenges faced by pregnant women. It's important to note that similar findings have been reported in a study conducted in Ethiopia [8,55].

In line with previous studies [56–58], our study findings indicate that pregnant women who didn't plan their pregnancy were associated with developing symptoms of CMD than their counterparts. An unplanned pregnancy can intensify stress and anxiety, resulting in emotional reactions such as fear, guilt, or ambivalence. This reaction may lead to emotional strain and affect mental health. Seeking support from family or mental health experts can help manage these conditions and reduce the risk of experiencing typical mental health issues during pregnancy [59,60].

In our study, pregnant women experiencing intimate partner violence were 2.2 times more likely to develop symptoms of common mental disorders than their counterparts, which is consistent with studies done in Ethiopia [8,15] and other countries [5,52,61,62]. Our Generalized Structural Equation Modeling (GSEM) analysis showed significant indirect effects of IPV on antenatal CMDs through social support. Social support acts as a buffer, modifying the adverse impact of IPV on maternal mental health by providing emotional support, thereby reducing stress and anxiety, enhancing coping strategies, and facilitating access to resources [63]. These findings underscore the importance of integrating social support interventions into prenatal care to protect and promote the mental health of pregnant women experiencing IPV. The potential of community-based programs, support groups, and educational campaigns to strengthen social support networks is immense. These initiatives can provide hope, ultimately improving maternal mental health outcomes [64,65].

In our finding, respondents who had poor social and partner support tended to have an increased risk of prenatal CMD compared to their counterparts, and our finding Is consistent with studies done in Ethiopia [19,66], China [67], and Australia [68]. Pregnant individuals lacking support from a partner or social network may experience heightened distress due to

the absence of a confidant to provide crucial information, guidance, or assistance. This phenomenon aligns with the buffering hypothesis, which suggests that social support can alleviate the adverse effects of stressful events[69,70]. This intervention effectively reduces mental distress by reshaping perceptions of adverse Events, reinforcing self-esteem, confidence, and self-efficacy, transferring coping resources, and enabling substantial changes in health-related behaviors [71,72].

In our Generalized Structural Equation Model (GSEM), we identified a significant mediating role of social support in the relationship between stress coping, antenatal common mental disorders, and partner support [73]. Our findings indicate that poor coping strategies for stress directly influence the risk of developing common antenatal mental disorders. This suggests that social support can strengthen individual coping resources by providing emotional assistance during pregnancy challenges; the reason might be that strategies such as seeking social support, problem-solving, and cognitive reappraisal are more accessible and effective when individuals have a strong support network to rely on [74]. A supportive relationship enhances individual resilience and adaptive coping skills, which are crucial for navigating the stressors associated with antenatal mental disorders. Additionally, it is noteworthy that partnered pregnant women in poor-quality relationships with their spouses face an elevated risk of common prenatal mental disorders due to increased stress and anxiety from their partners [75]. Moreover, evidence suggests that a lack of partner support is associated with increased prenatal stress among low-income women [76].

Our study found that experiencing life-threatening events is significantly linked to common mental disorders in pregnant women. Stressful events directly impact antenatal mental disorders and may affect pregnant women due to various biological, psychological, and social factors. Stressful events can worsen vulnerability to mental health issues during pregnancy by triggering physiological stress responses, disrupting mood regulation, and worsening psychosocial stressors [77,78].

Our research, consistent with previous studies conducted in Haromaya [8] and Gondar [79], Ethiopia, highlights the potential benefits of ANC follow-up. We found that pregnant women who did not have ANC follow-up for their current pregnancy were more likely to have CMD than those who had ANC visits. This suggests that antenatal clinic attendance could be a powerful tool in constructing maternal self-esteem and resiliency, increasing the chance of obtaining information about pregnancy preparedness and minimizing risk factors.

In light of the findings discussed future research could benefit from exploring additional mediators, such as examining the role of economic stability and mental health education. This could provide insights into how financial security impacts mental health outcomes during pregnancy and help develop targeted interventions. Additionally, examining the availability of mental health education is vital, as it can identify knowledge gaps that may lead to poor mental health outcomes.

## Study limitations and strengths

Our study holds significant implications for advancing maternal mental health promotion and prevention efforts. We utilized the stress process theoretical model and conducted a comprehensive generalized structural equation modeling analysis to thoroughly investigate the diverse stressors and pathways that underlie common antenatal mental disorders. The insights from this analysis offer valuable contributions to the existing literature on this subject matter. The observations made by our study are indeed of great significance; however, it is essential to acknowledge that they do not inherently imply causation due to the inherent limitations of our study design. While the findings are substantial, believing the study's design limitations and the impact of the scales' reliability are commendable. It highlights the importance of

methodological rigor and the need for longitudinal studies to ascertain causation. The call for improved validation of social support and stress-coping scales is crucial, as it will enhance the integrity of future research. This study serves as a stepping stone for further exploration and underscores the necessity of continuous improvement in research methodologies to understand better and address maternal mental health issues.

## Conclusion

The research underscores the complex relationship between different stressors and prevalent antenatal mental disorders. The study reveals that the absence of prenatal care initiation, a familial history of mental health conditions, experiences of intimate partner violence, and encountering an unplanned pregnancy significantly contribute to adverse impacts on the mental well-being of pregnant mothers. Furthermore, these stressors can indirectly influence mental health outcomes through the mediation of stressful life events, stress coping mechanisms and social and partner support. As a result, targeted interventions focused on enhancing coping resources, strengthening social support systems, and addressing underlying psychosocial stressors demonstrate efficacy in improving the risks associated with prevalent antenatal mental disorders, thus contributing to favorable maternal and fetal outcomes. Additionally, enhancing training for healthcare workers and increasing mothers' awareness of mental health coping strategies during pregnancy is crucial for improving mental health outcomes. By integrating mental health education into prenatal care, establishing peer support networks, and leveraging technology for accessible resources, we can empower both healthcare providers and mothers. Community outreach and collaborative research will further enhance this approach, fostering resilience and contributing to healthier families and communities.

These results have significant implications for healthcare professionals and policymakers, providing them with valuable insights into the determinants of maternal mental health during pregnancy and facilitating the implementation of practical measures to provide pregnant women with comprehensive support and care.

## Supporting information

**S1 Table. Bivariable and Multivariable analysis of antenatal common mental disorders among pregnant women in Dabat HDSS Northwest, Ethiopia.**
(XLSX)

## Acknowledgments

We want to express our gratitude to the individuals who made this study possible, including the participants, data collectors, managers, supervisors, and staff of the Dabat HDSS office who facilitated the data collection process.

## Author contributions

**Conceptualization:** Helina Abebe Kurbi, Solomon Mekonnen Abebe, Netsanet Worku Mengistu, Alemayehu Teklu Toni, Tadesse Awoke Ayele.

**Data curation:** Helina Abebe Kurbi.

**Formal analysis:** Helina Abebe Kurbi, Solomon Mekonnen Abebe, Tadesse Awoke Ayele.

**Methodology:** Helina Abebe Kurbi, Solomon Mekonnen Abebe, Netsanet Worku Mengistu, Tadesse Awoke Ayele.

**Project administration:** Helina Abebe Kurbi.

**Supervision:** Solomon Mekonnen Abebe, Netsanet Worku Mengistu, Alemayehu Teklu Toni, Tadesse Awoke Ayele.

**Validation:** Solomon Mekonnen Abebe, Netsanet Worku Mengistu, Alemayehu Teklu Toni, Tadesse Awoke Ayele.

**Visualization:** Helina Abebe Kurbi, Solomon Mekonnen Abebe, Tadesse Awoke Ayele.

**Writing – original draft:** Helina Abebe Kurbi.

**Writing – review & editing:** Helina Abebe Kurbi, Solomon Mekonnen Abebe, Netsanet Worku Mengistu, Alemayehu Teklu Toni, Tadesse Awoke Ayele.

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
