## [Decision Letter · Decision Letter 0]

26 Nov 2024

PMEN-D-24-00327

The Causal Effect of Stressors on Antenatal Common Mental Disorders Among Pregnant Mothers in Dabat District Northwest, Ethiopia: A Generalized Structural Equation Model Approach.

PLOS Mental Health

Dear Dr. Kurbi,

Thank you for submitting your manuscript to PLOS Mental Health. We apologise for the severe delay in reaching a decision and appreciate that this would have been frustrating. Thank you for your patience. After careful consideration of the reviewer reports, we feel that your manuscript has merit but does not fully meet PLOS Mental Health’s publication criteria as it currently stands. Therefore, we invite you to submit a revised version of the manuscript that addresses the points raised during the review process.

Please fully address all concerns raised by the reviewers.

We look forward to receiving your revised manuscript.

Kind regards,

Karli Montague-Cardoso

Executive Editor

PLOS Mental Health

Journal Requirements:

https://journals.plos.org/mentalhealth/s/figures 

https://journals.plos.org/mentalhealth/s/figures#loc-file-requirements

Additional Editor Comments (if provided):

Reviewers' comments:

Reviewer's Responses to Questions

**Comments to the Author**

1. Does this manuscript meet PLOS Mental Health’s publication criteria? Is the manuscript technically sound, and do the data support the conclusions? The manuscript must describe methodologically and ethically rigorous research with conclusions that are appropriately drawn based on the data presented.

Reviewer #1: Partly

Reviewer #2: Yes

2. Has the statistical analysis been performed appropriately and rigorously?

Reviewer #1: No

Reviewer #2: Yes

3. Have the authors made all data underlying the findings in their manuscript fully available (please refer to the Data Availability Statement at the start of the manuscript PDF file)?

Reviewer #1: No

Reviewer #2: Yes

4. Is the manuscript presented in an intelligible fashion and written in standard English?

Reviewer #1: Yes

Reviewer #2: Yes

5. Review Comments to the Author

Reviewer #1: Thanks, authors for presenting the perspective of the prevalence and factors associated with Common Mental Disorders among pregnant women in Ethiopia.

Below are the comments to improve the manuscript.

Introduction:

- Better to add more about the global context to the Ethiopian context

- It would be better if you clarify the statement “prevalence of common mental disorders (CMD) among pregnant mothers that ranged from 12% to 37% ” whether it is local or national data.

- It is better to add the rationale for the prevalence of CMD

Methods:

- Explain why the authors have only included pregnant women in the second and third trimester since the prevalence of CMD varies across trimesters.

- How does the ODK software support the data accuracy?

- The authors should explain how the multicollinearity was handled.

Result:

- Better to have a row percentage for the comparison in Tables 1 and 2, similar in figure 2 as well

Discussion:

- It would be good if the authors suggest future research, such as other mediators like economic stability or availability of mental health education that might want to be considered to further clarify antenatal mental health dynamics.

Conclusion:

- Can suggest further works in the training of healthcare workers and mothers' awareness on coping strategies for mental health.

Reviewer #2: 1. If this was a cross-sectional study, what does a ratio of 1:5 mean in the sampling? It is not evident in the results that the distribution of subjects is 1:5 based on exposure to CMD

2. What was the modal parity and mean gestational age of the subjects?

3. Figure 2 is incomplete

4. There are spelling mistakes in the tables

5. See text for other corrections

6. PLOS authors have the option to publish the peer review history of their article (what does this mean?). If published, this will include your full peer review and any attached files.

**Do you want your identity to be public for this peer review?** For information about this choice, including consent withdrawal, please see our Privacy Policy.

Reviewer #1: No

Reviewer #2: **Yes: **Dr Bilal Sulaiman

---

## [Decision Letter · Decision Letter 1]

21 Jan 2025

The Causal Effect of Stressors on Antenatal Common Mental Disorders Among Pregnant Mothers in Dabat District Northwest, Ethiopia: A Generalized Structural Equation Model Approach.

PMEN-D-24-00327R1

Dear Researcher, mental health specialist Kurbi,

We are pleased to inform you that your manuscript 'The Causal Effect of Stressors on Antenatal Common Mental Disorders Among Pregnant Mothers in Dabat District Northwest, Ethiopia: A Generalized Structural Equation Model Approach.' has been provisionally accepted for publication in PLOS Mental Health.

Best regards,

Karli Montague-Cardoso

Executive Editor

PLOS Mental Health

Reviewer Comments (if any, and for reference):

Reviewer's Responses to Questions

**Comments to the Author**

1. If the authors have adequately addressed your comments raised in a previous round of review and you feel that this manuscript is now acceptable for publication, you may indicate that here to bypass the “Comments to the Author” section, enter your conflict of interest statement in the “Confidential to Editor” section, and submit your "Accept" recommendation.

Reviewer #1: All comments have been addressed

2. Does this manuscript meet PLOS Mental Health’s publication criteria? Is the manuscript technically sound, and do the data support the conclusions? The manuscript must describe methodologically and ethically rigorous research with conclusions that are appropriately drawn based on the data presented.

Reviewer #1: Yes

3. Has the statistical analysis been performed appropriately and rigorously?

Reviewer #1: Yes

4. Have the authors made all data underlying the findings in their manuscript fully available (please refer to the Data Availability Statement at the start of the manuscript PDF file)?

Reviewer #1: No

5. Is the manuscript presented in an intelligible fashion and written in standard English?

Reviewer #1: Yes

6. Review Comments to the Author

Reviewer #1: Thank you for addressing the suggestions. It is better to keep and interpret the row percentage in table 2 rather including both row and column percentages.

7. PLOS authors have the option to publish the peer review history of their article (what does this mean?). If published, this will include your full peer review and any attached files.

**Do you want your identity to be public for this peer review?** For information about this choice, including consent withdrawal, please see our Privacy Policy.

Reviewer #1: No
